# A 4 × 4 Active Antenna Array with Adjustable Beam Steering

**DOI:** 10.3390/s23031324

**Published:** 2023-01-24

**Authors:** Sebastian Verho, Van Thang Nguyen, Jae-Young Chung

**Affiliations:** Department of Electrical and Information Engineering, Seoul National University of Science and Technology, Seoul 01811, Republic of Korea

**Keywords:** antenna array, beam steering, polarization control, S-band

## Abstract

An adjustable 4×4 antenna array with electrical beam steering and polarization control is presented. Here, adjustability means the ability to correct the beam steering angle post-calibration. The objective is to improve the steering accuracy which is critical in point-to-point communication as inaccuracy will cause transmission failure due to a missed target. The accuracy is enhanced by adjusting the beam steering angle in beamforming calculations. To execute this, the system is calibrated by measuring several unit cells of a partial 4×4 array structure at different voltage bias points and calculating an average model of the phase shift profile. This reduces the phase error from variations between components and robust beam steering is achieved. This technique is utilized in far-field measurements, and fairly accurate initial beam steering angles are achieved at 3 GHz. The accuracy is further improved by over or under steering the desired angle in the beamforming calculations to finally achieve the steering angle of interest with an accuracy of 2${}^{\circ }$. Overall, the main beam is incrementally steered from $0^{\circ }$ to $45^{\circ }$ with the gain ranging from 4.7 dB to 2.8 dB. The polarization control is also demonstrated in horizontal and vertical directions for a linearly polarized wave.

## 1. Introduction

Antenna arrays are a popular choice for point-to-point wireless communications due to their ability to concentrate radiated power enabling high directivity [1]. To increase their functionality, they can also be made reconfigurable. Reconfigurability is a broad term which can refer to several concepts, e.g., electrical beam steering, polarization control, pattern diversity, beamforming, beamsplitting, and frequency reconfigurability [2,3,4,5]. Examples of typical reconfigurable antenna array systems are phased arrays, transmitarrays/reflectarrays, metasurfaces, and frequency selective surfaces (FSS). In recent years, reconfigurable intelligent surfaces (RIS) have garnered more interest and offer reconfigurable properties similar to those in the other aforementioned systems as well as this work [6,7,8]. RIS can be considered as a reflectarray and metasurface hybrid [9]. The main difference with RIS is its distance to the transmitting source and receiver as it is typically located between the two to create a smart radio environment which enhances the signal propagation between the two points [10,11]. For future antenna array systems, hotspot-mediated control [12] shows promise in being the next possible step in antenna array development as it provides reconfigurability through plasmonic resonances instead of semiconductor components. This removes the frequency limitation imposed by the said components and also enables the nanoscale antenna designs.

Among the above reconfigurable properties, this paper focuses on the electrical beam steering and also presents some results for polarization control. The former is a highly desired property as it removes or reduces the need to move a system mechanically. Furthermore, precise beam steering is critical in long-distance communications as a few degrees of variation in the steering angle can lead to the transmitted signal missing the receiver and causing transmission failure in highly directive systems, i.e., narrow main beam. Practically, beam steering can be implemented by embedding active components, e.g., PIN diodes, varactors, microelectromechanical systems (MEMS), and monolithic microwave integrated circuits (MMIC), to patterns created on printed circuit boards (PCB). Another option is materials whose material properties are altered with a voltage bias, such as a liquid crystal and graphene.

As mentioned in [13], PIN and varactor diodes are preferred to radio frequency (RF) and MEMS switches due to their lower cost and higher reliability. Therefore, the design of this paper also employs PIN and varactor diodes to achieve reconfigurability. The PIN diodes are used to implement polarization control and the varactors are utilized in beam steering.

Even though this paper and other works utilized varactors [14,15,16,17] for implementing beam steering, it was also accomplished with PIN diodes, as was demonstrated in [18,19,20,21,22]. Interestingly, a combination of both components was utilized in [23,24,25]. With regard to beam steering, the main difference between these two components is the phase profile they provide. PIN diodes are discrete devices meaning that they are used in ON or OFF states, i.e., two phase states. Varactors on the other hand can be considered as tunable capacitances and provide continuous phase control. However, references [22,24] demonstrated that the continuous phase can also be discretized. Nevertheless, these are important factors to consider since, in order to perform the aforementioned beam steering, it is required to generate a progressive phase difference between the antenna elements as was discussed, for example, in [13]. If the phase is not precisely generated, it introduces a phase error. With PIN diodes, this phase error is commonly called a quantization error due to its discrete nature. The effect of the quantization error was studied in [26] where the authors reported that the phase error generated strong side lobes and a grating lobe. Additionally, an increase in side lobe levels (SLL) was also observed. More importantly, the beam steering became less accurate and the main lobe gain degraded especially at large steering angles, i.e., high scan loss. Therefore, implementations with PIN diodes have an inherent disadvantage compared to varactors. However, the strength of the PIN diode is in its simple characterization as it only has the two states. For the varactor, the characterization is not only difficult due to its continuous phase profile but it is also strongly nonlinear. In the other works using varactors for beam steering [14,15,16,17,23,24,25], there was little discussion about the characterization. Instead, references [14,23,25] only mentioned the capacitance range of the varactor which is not sufficient to create an accurate phase profile because inductive components also affect the phase, as mentioned in [17]. These works successfully performed the beam steering but there was a lack of discussion about its accuracy. Furthermore, looking at Figure 15 in [14], it was visually confirmed that some steering angles do not have their peaks exactly at the claimed angles. This appears to have been rectified, but not discussed in the following work [23], using the same varactor.

Reis et al. [17] characterized the phase of a single FSS unit cell for beam steering. However, in the far-field measurements, a maximum steering error of 16${}^{\circ }$ was reported at the steering angle of 45${}^{\circ }$ in the azimuth direction. The authors mentioned the need to compensate for angle mismatches in angles above 15${}^{\circ }$ with lookup tables. Nevertheless, looking at the gains and SLLs, there is strong variation in these values between the different steering angles hinting towards a strong phase error, as per earlier discussion.

Improved results can be achieved by characterizing each unit cell individually as was performed in Lau et al. [15]. However, the characterization method was not mentioned. Additionally, the steering accuracy was not discussed and it was visually confirmed that the steering angles of >$\pm 40^{\circ }$ in Figures 13–16 are not exactly at the claimed angles. There were also gain variations between different steering angles.

Similarly to the above, Nicholls et al. [16] characterized each unit cell individually. This was performed with a near-field probe and the information of each unit cell’s phase characteristics was added to a lookup table. Moreover, they provided exhaustive data about the accuracy of each steering angle. The main beam was scanned in the E- and H-planes as well as diagonally. For example, in the E-plane, the maximum steering error was 1.9${}^{\circ }$ and 6.6${}^{\circ }$ in theta and phi directions, respectively. From this, it is evident that even though each unit cell was accurately characterized, accurate beam steering was not a trivial task and there is bound to be some error, even with careful characterization.

Considering the above points, this work proposes a characterization setup which only contains the phase shifters and feeding network of the 4×4 antenna array design of this paper. The input and 16 outputs are terminated with SubMiniature version A (SMA) connectors in order to measure each unit cell individually with a vector network analyzer (VNA). With these measurements, the operation of the phase shifters and feeding network is confirmed. Moreover, the total phase shift of each unit cell is measured and the results are used to form an average model of the phase shift profile. This is not expected to provide an optimal performance in the beam steering as the objective is to create a single fairly precise model used as a baseline and tolerant to component and fabrication variations. The authors want to emphasize that even though each unit cell is measured individually, these results cannot be used as-is in the 4×4 antenna array. The reason for this is that the characterization board is not the full system (4×4 antenna array) used in far-field measurements. Therefore, there will be small variations in the etched board line widths and components between the two boards which causes some discrepancy. Hence, the average phase shift model is used instead.

The model is used in far-field measurements to steer the main radiated beam. If there is a discrepancy between the target and measured angle, the beam direction is adjusted by over or under steering the beam in the beamforming calculations until the target angle is reached within an acceptable angle error. The benefit of this approach is that the full antenna array can be pre-calibrated offsite before performing any far-field measurements and the final adjustments are performed later during the radiation pattern measurements. This can be a very beneficial approach in situations where the limited measurement time in an anechoic chamber is available or there is some fault in the board requiring readjustment. This kind of adjustability is not commonly discussed in related works to the best of the authors’ knowledge. As demonstrated later, the approach of this paper is simple and easy-to-use as it requires only changing the steering angle value in the beamforming calculations.

In order to demonstrate the feasibility of the above characterization method, the design and measurement results of a 4×4 antenna array with electrical beam steering are presented. In addition, the array is also capable of performing polarization control, which will also be shown. The results are analyzed as well as compared to other similar works to highlight the strengths of the approach we employ.

Then, we discuss the contributions of this work. First, we provide a method to analytically calculate the phase shift of each unit cell for beam steering. This does not require the individual characterization of each unit cell as in [15,16]. Instead, we create an average model that offers robust beam steering performance. Due to this approach, the beam steering is also adjustable by modifying the steering angle in beamforming calculations, making it simple to re-adjust. Additionally, the method presented herein only requires a VNA instead of a near-field probe. Therefore, this characterization does not require radiation measurement equipment for calibration. Hence, the system can be pre-calibrated and re-adjusted for far-field measurements meaning that this method requires less measurement time in an anechoic chamber. Finally, we also provide insight into different loss sources such as the surface finish and surface roughness through a comparison between simulations and measurements which is not commonly discussed in related works.

This paper is organized as follows. First, the complete structure and its general details are introduced. After this, the structure is broken down into smaller sections followed by the introduction of the power divider and phase shifter network. Then, a more detailed analysis of the phase shifter and the phase shifting is provided from where we move to the antenna. In this section, the polarization control is presented. This concludes the theory part and we proceed to the measurement results. The first measurement setup is the phase shift characterization with a board containing only the power divider and phase shifter network. From this, the full structure far-field measurement results and an analysis are presented. The results are compared to other works after this. Lastly, a conclusion is provided.

## 2. Complete Structure

The 4×4 antenna array structure is shown in Figure 1. It consists of three main designs: power divider, phase shifter, and antenna. The power divider and phase shifter were designed on a Taconic RF-10 substrate ($\varepsilon_{r}$ = 10.2, tan δ = 0.0025) with a thickness of 0.635 mm. For the antenna design, Taconic TRF-43 substrate is utilized. The thickness is chosen as 3.175 mm and the material properties are $\varepsilon_{r}$ = 4.3, and tan δ = 0.0035. Additionally, there is a 0.1 mm thick bonding layer, Isola 185HR, to co-join the two substrates together.

Input power is fed through an SMA connector which is connected to the power divider feed network. Four levels of power dividers are cascaded to split the power equally among the 16 unit cells. Following the power divider network are the phase shifters, as illustrated in Figure 2. Each unit cell has a two-stage phase shifter which is designed to provide over $360^{\circ }$ of phase shift. They are controlled independently from each other with a direct current (DC) bias voltage. The phase shifters are connected to the antennas with metallized vias.

The antennas in Figure 1 are spaced $0.4\lambda_{0}\times 0.4\lambda_{0}$ from each other equaling to 40 mm at the operation frequency of 3 GHz. Hence, the total aperture size without bias lines or connectors is 160 mm × 160 mm.

### 2.1. Power Divider

A Wilkinson power divider is employed for the power division. A ring structure is used for the 70.71 Ω section to reduce its footprint. A 100 Ω resistor is embedded into the ring, completing the structure. In total, there are four levels of power division. Therefore, the ideal power at each output is −12 dB.

The input and output of the power divider are initially designed as 50 Ω and optimized with Ansys HFSS to provide an equal power division to the unit cells while having a low reflection loss. Its simulation results are shown in Figure 3b. The average output is −12.8 dB, meaning that there is 0.8 dB of loss. The $S_{11}$ is −13.0 dB with a −10 dB bandwidth of approximately 20%.

### 2.2. Phase Shifter

A reflective type phase shifter (RTPS) was chosen as the phase shifter design. As mentioned previously, a two-stage structure is used to ensure $360^{\circ }$ phase shift when altering the voltage from 0 to 12 V. As mentioned in [27,28], it is necessary to have at least a total of four varactors to fulfill this condition due to the limited capacitance ratio of the varactors.

In a single stage, a four-port $90^{\circ }$ quadrature coupler is used to split the input power into two reflective loads. A single load consists of a varactor and a via-terminated microstrip line. The name reflective load is used here because the phase shifter operation is based on reflection. The well-known equation for reflection is
(1)$$\Gamma =\frac{Z_{0}-Z_{\mathrm{load}}}{Z_{0}-Z_{\mathrm{load}}},$$
where $Z_{0}$ is the system impedance and $Z_{\mathrm{load}}$ is the reflective load impedance [28].

The varactor can be modeled with an equivalent circuit consisting of resistor-capacitor-inductor (RLC) components as shown in Figure 4.

$C_{j}$ is the junction capacitance whereas $L_{S}$ and $C_{p}$ are parasitics due to the package of the varactor. $R_{S}$ represents the losses in the component [29]. The model can be simplified by combining the package capacitance $C_{p}$ with the junction capacitance $C_{j}$ to form $C_{v}$ [27]. This simplifies the simulation model of a varactor since only one RLC boundary is required.

Macom MAVR-000120-14110 was chosen as the varactor which provides a variable capacitance with a capacitance ratio $C_{\mathrm{ratio}}$ of 5.5 [30]. It is controlled with a reverse DC bias voltage. This provides tunability to the structure enabling relative phase shift control. The relation between the voltage and capacitance is expressed with
(2)$$C_{v}\left(V_{\mathrm{bias}}\right)=\frac{C_{j}}{{\left(1+\frac{V_{j}}{V_{\mathrm{bias}}}\right)}^{m}}.$$

In (Equation 2), $C_{v}$ is the total capacitance, $V_{\mathrm{bias}}$ is the bias voltage, $C_{j}$ is the junction capacitance of the varactor, $V_{j}$ is the junction potential, and *m* is the grading coefficient [29].

The via-terminated microstrip line mainly functions as a fixed inductance and also provides a ground connection for the varactor DC biasing. Together with the varactor capacitance, they provide the necessary reactive elements to obtain over $360^{\circ }$ of the phase shift. The phase shift can be decreased or increased by making the microstrip line shorter or longer, i.e., decreasing or increasing the inductance. The total phase shift is expressed with
(3)$$∠S_{21}=-90^{\circ }-2\arctan\left(\frac{X_{\mathrm{load}}}{Z_{0}}\right).$$

$X_{\mathrm{load}}$ represents the total reactance [31]. $X_{\mathrm{load}}$ is decomposed into
(4)$$X_{\mathrm{load}}=X_{L}-X_{C}=2\pi fL_{\mathrm{tot}}-\frac{1}{2\pi fC_{v}},$$
where *f* is the operating frequency and $L_{\mathrm{tot}}$ is the total inductance of the varactor and transmission line.

It is worth noting that there is a limit on the increase in phase shift with the increase in the microstrip line length. This happens when the inductive reactance $X_{L}$ in (Equation 4) becomes much larger than the capacitive reactance $X_{C}$. In this case, an additional increase in inductance will reduce the total phase shift. This is because changing the capacitance has a smaller effect on the overall reactance, i.e., the inductance saturates the total reactance. More information about the effect of series and parallel components in a reflective load can be found in [32].

A detailed depiction of the designed phase shifter is presented in Figure 5. There are several RLC components. First, a 15 kΩ resistor is added to prevent the RF signal leaking to the bias line meant for varactor control. The varactors are kept in a reverse bias state. Therefore, they draw little to no DC current. Consequently, using a resistor does not significantly increase DC power loss. Then, to ensure that the DC control signal of a phase shifter does not interfere with the antenna components or other phase shifters, a 51 pF capacitor is inserted into the input and output of the phase shifter, respectively. Finally, a 4.7 nH inductor was added between the ground and microstrip line to provide a DC ground for the antenna side components.

The structure of Figure 5 is simulated from 2.0 GHz to 4.0 GHz and the varactor voltage is swept from 0 V to 12 V. To model the varactor, (Equation 4) is used with values from the datasheet [30]. The $S_{11}$ and $S_{21}$ results are presented in Figure 6. The average $S_{11}$ and $S_{21}$ at 3 GHz are −11.1 dB and −1.7 dB, respectively. The total phase shift is 437${}^{\circ }$. This is an initial model that will later be updated accordingly in Section 3 based on measurement results.

### 2.3. Antenna

The patch antenna from [14] was modified to fit the purpose of this design. The structure is shown in Figure 7a. In addition to operating as an antenna, the polarization control is also implemented on it. This is performed by switching the two PIN diodes, PIN Diode 1 and PIN Diode 2, ON and OFF in a complimentary manner. For example, if PIN Diode 1 is ON and PIN Diode 2 is OFF, the antenna operates in horizontal polarization mode, and if the switching is performed the other way around, then vertical polarization mode is activated.

To achieve the individual control of said diodes, two bias lines are added to feed DC. Each bias line has a 270 nH inductor to block the RF signal from leaking to the bias line. Additionally, a 100 Ω resistor is added to protect the PIN diodes from transient condition current spikes and to ensure that the current is evenly distributed between several parallel PIN diodes sharing the same bias line in an antenna array configuration.

To prevent the mixing of the DC signals, multiple gaps were added to the structure. The RF signal can pass through the gaps, but for DC signals, they are open connections. The effect of the gaps was carefully considered to keep the patch antenna operating at 3 GHz and to ensure high cross polar discrimination (XPD). The initial design had a diagonal gap at the top right and bottom left corner of the O-shape structure, respectively. Although this approach minimized the number of gaps, the XPD was low because the surface current did not flow along the desired path, i.e., the two polarization modes were not distinct. To achieve a high XPD, the bottom left gap was split into two gaps, as illustrated in Figure 7a.

Thanks to the individual control of the diodes, a high reverse voltage can be applied to the OFF state PIN diode. This in theory improves isolation and ensures that the accidental turning ON is avoided. Moreover, high XPD is guaranteed. However, this is not demonstrated in this paper due the limitation of the control board used in measurements.

Macom MA4AGP907 was chosen as the PIN diode for the polarization control. The equivalent circuits shown in Figure 7b represent the ON and OFF state, respectively. Based on the datasheet [33], the parameters for the ON state were chosen as $R_{\mathrm{ON}}=5.2\;\Omega$ and $L_{\mathrm{ON}}=0.6\;\mathrm{nH}$. For the OFF state, the parameters are $C_{\mathrm{OFF}}=0.025\;\mathrm{pF}$ and $L_{\mathrm{OFF}}=0.6\;\mathrm{nH}$.

The antenna was simulated in HFSS in both horizontal and vertical polarization states. The results of the broadside radiation pattern at $\phi =90^{\circ }$ are shown in Figure 8. When the antenna was set to the horizontal polarization state, the $S_{11}$ at 3 GHz was −20.2 dB with a fractional bandwidth of 4.0%. The peak gain was 3.2 dBi. In the vertical polarization state, the $S_{11}$ was −21.9 dB with a fractional bandwidth of 4.1% and a gain of 3.5 dBi. Both polarization states have a XPD of over 30 dB.

## 3. Phase Shift Characterization

In order to determine the overall phase shift of the 4×4 structure, a board containing only the power divider and phase shifter was fabricated, shown in Figure 9. An SMA connector was soldered to each output of the 16 unit cells as is displayed in Figure 9b. The unit cells were measured one by one with an Anritsu MS46122B VNA by adjusting the respective unit cell’s bias voltage for varactors from 0 V to 12 V in 1 V steps. The other unit cells were terminated with a 50 Ω load. The voltage control was performed with a third-party control board that was custom made for this work based on specifications provided by us. The measurement was repeated with an identical board to collect sufficient data for an average model of the phase shift at several bias points. The measurement setup is shown in Figure 10.

The measured total average total phase shift was $462^{\circ }$ with a $29^{\circ }$ difference between the highest and lowest total phase shift. The average phase shift is $25^{\circ }$ higher than in the simulation. This is believed to be due to the additional inductance from the solder that was not considered before. The differences in phase shift between the unit cells is assumed to be due to variations in varactors. For example, in the varactor datasheet [30], the minimum and maximum capacitances are 0.30 pF and 0.40 pF at 4 V, respectively.

$S_{11}$ and $S_{21}$ results are shown in Figure 11. In the legend, codes representing each unit cell are shown. The position of each unit cell is mapped in Figure 11c which represents the phase shifter and power divider layer illustrated in Figure 2. The average $S_{11}$ and $S_{21}$ were −10.9 dB and −18.9 dB, respectively. Ideally, the $S_{21}$ would be −12 dB, however, due to the losses in the power dividers, phase shifters (imperfect reflection), and non-ideal conductors, the electroless nickel immersion (ENIG) surface finish and surface roughness, there was 6.9 dB of additional loss on average. These surface finish and surface roughness losses are discussed in greater detail in Section 4. The variation in loss between different bias voltages is due to the changing reflection coefficient of the RTPS. The highest loss is at 1 V and, in this case, the series inductance and capacitance are in resonance, i.e., (Equation 4) is close to zero. In this case, the reflection coefficient is purely based on the internal resistance of the varactor ($R_{s}$ in Figure 4). Additionally, imbalanced power division was observed for unit cells near the feed of the structure, i.e., B21, B22, B31, and B32. This was also observed in the simulation results of Figure 3, however, the difference between the highest and lowest loss was only 0.2 dB. The higher losses caused by the non-ideal conductors, ENIG, and surface roughness are believed to have increased this difference even more.

To obtain an updated varactor model with the measurement data, a curve fit was performed with MATLAB’s built-in curve fit toolbox by tuning the parameters of (Equation 2) with the minimum square error (MMSE) technique. While tuning, the phase shift at different bias points was calculated with (Equation 3) and compared to the measurement data. This was iterated until the error between the calculations and measurement results was minimized. The result of the curve fit is in Figure 12, showing good agreement with the measured data. The varactor model parameters obtained from the curve fit are presented in Table 1. Although these values emulate the phase profile well, they are not expected to be accurate in terms of inductance and capacitance. Therefore, a frequency shift in simulation results is expected when compared to measurements. In future work, the goal is to improve this method to accurately capture the inductance and capacitance values as well. Nevertheless, it is sufficient for beam steering purposes as will be demonstrated in Section 4.

### 3.1. Error Analysis

To understand the effect of the average phase shift model to phase error, four different measurement scenarios were performed. Each scenario represents a different beam steering angle of 10${}^{\circ }$, 20${}^{\circ }$, 30${}^{\circ }$, and 40${}^{\circ }$, respectively. Based on calculations, a different voltage was applied to each column of unit cells in the board. More details about these calculations are provided in Section 4.1.

The results are shown for 10${}^{\circ }$ and 40${}^{\circ }$ cases in Table 2 and Table 3, respectively. The bottom two rows of the board, marked as row 3 and row 4, were measured and compared to each other. The phases are normalized in relation to the first unit cell of each row. Both rows have identical voltages applied to their respective unit cell’s phase shifters. The voltages were confirmed to be correct with a multimeter with a maximum error of 0.05 V. Ideally, this kind of control would generate equal phase shifts at the output of each unit cell of a column. However, there is a discrepancy in the phase shifts due to variations in varactors, as discussed earlier. For the 10${}^{\circ }$ case in Table 2, the maximum phase error is −4.6${}^{\circ }$ at the unit cell B42 of row 4 which is greater than the maximum error of Row 3, 1.7${}^{\circ }$.

For the 40${}^{\circ }$ case in Table 3, the amount of error increases since a wider voltage range is utilized. The maximum error is observed at B31 of Row 4 equaling to 16.3${}^{\circ }$. The maximum error between the rows is 6.1${}^{\circ }$, which is a fairly good result considering the use of the average model instead of individual characterization.

Overall, the worst error observed compared to calculations was for the 30${}^{\circ }$ case at B33 which was 22.7${}^{\circ }$. As for the error between rows, the worst case was at a 20${}^{\circ }$ steering angle with an error of 6.5${}^{\circ }$. Since the phase error between the rows is small, the radiated wave fronts of the full 4×4 antenna array will not cancel each other destructively. Therefore, it is expected that the use of the average model will not significantly degrade the gain of the full 4×4 antenna array in beam steering. However, the steering angle will not likely be exactly at the expected angle due to the error between the calculated and measured phase value which will require some adjustment in the beamforming calculations.

## 4. Far-Field Measurements

For the far-field measurement, the 4×4 antenna array shown in Figure 13 was fabricated. The total size of it is 420 mm × 555 mm and it contains a DC connector shown in Figure 13b. It is connected to the varactor and PIN diode bias lines. As in the previous section, the components are controlled with the control board.

### 4.1. Beamforming Strategy

For the beam steering, the beamforming method introduced in [13] was used. This has also been used in [15,17], for example. The progressive phase for each unit cell for a 2D array is calculated with the following equation
(5)$$\left\{\begin{array}{c} \psi_{x}=-k_{0}p\sin\left(\theta \right)\cos\left(\phi \right)\\ \psi_{y}=-k_{0}p\sin\left(\theta \right)\sin\left(\phi \right). \end{array}\right.$$

The $\psi_{x}$ and $\psi_{y}$ are the progressive phases for the X and Y directions, respectively. $k_{0}=2\pi /\lambda_{0}$ is the free space wave number, *p* is the periodicity of unit cells, θ and ϕ are the steering angles in spherical coordinates.

With (Equation 5), the progressive phase shift is calculated for each unit cell. Then, with (Equation 2), (Equation 3), and (Equation 4), the equivalent voltage value to produce the said phase is calculated. All of these calculations are performed with a MATLAB script written for this application. The calculated voltage values are input into a program that controls the control board.

Based upon the results shown in Figure 11b, 12 V is used as the base voltage for calculating the voltages since lowest insertion loss was observed at that value in Section 3. All other phases are calculated and normalized in relation to the phase of 12 V.

### 4.2. Measurement Setup

The measurement was performed in an anechoic chamber shown in Figure 14. The control board was connected to a power supply. These were placed along the laptop on a platform which rotates 360${}^{\circ }$ on each 2D scan. The communication protocol between the laptop and control board is serial-to-USB. The 4×4 antenna array was connected to the control board with 2 m-long cables. The transmitted power was measured with a horn antenna that can be rotated 90${}^{\circ }$ in order to measure the different polarization states providing the cross-polarization levels as well. The measuring resolution of this setup is 1${}^{\circ }$.

### 4.3. Measurement Results

The goal was to measure electrical beam steering from 0${}^{\circ }$ to 45${}^{\circ }$ in 5${}^{\circ }$ steps. The antennas were set to horizontal polarization in each measurement. The results are presented in Figure 15. The observed maximum and minimum gains were 4.7 dBi and 2.8 dBi, respectively. The variation can be partly explained with scan loss. Another reason is the phase error which was discussed in Section 3.1. In addition, the unequal insertion loss of the phase shifter at different bias voltages is also affected since, during beam steering, each antenna column has a different voltage applied their respective phase shifters to generate the required progressive phase shift. The phase error and unequal loss also negatively affect the SLL. The peak gain and SLL for each steering angle are shown in Table 4. The gain for 0${}^{\circ }$ was lower than at 5${}^{\circ }$ because the 0${}^{\circ }$ control was accidentally set to 10 V to each unit cell instead of the more optimal 12 V.

The control angle in Table 4 is the angle used in beamforming calculations to achieve the target angle with a maximum error of 2${}^{\circ }$. These were obtained by initially attempting to steer to 10${}^{\circ }$, 20${}^{\circ }$, 30${}^{\circ }$, 40${}^{\circ }$, and 45${}^{\circ }$, i.e., setting the control angle to these values, and then the control angles were adjusted until the beam steering was achieved at every target angle. As seen from the results, the minimum and maximum target angles were very close to the control angles, meaning that the system is fairly well calibrated. However, the angles in-between needed some tuning and the target angle of 25${}^{\circ }$ had the highest discrepancy of 7${}^{\circ }$. As mentioned in Section 3.1, the steering angles 20${}^{\circ }$ and 30${}^{\circ }$ had the largest errors agreeing with this result. Nevertheless, this result demonstrates the easy adjustability of the 4×4 antenna array while having a low scan loss (1.9 dB), i.e., a low phase error.

To demonstrate the polarization control, the antenna array was set to a 0${}^{\circ }$ steering angle and each state was measured separately. For horizontal polarization control, the horizontal direction PIN diodes (PIN Diode 1 in Figure 7) of each antenna were set to ON state and the voltage over them was adjusted so each diode conducts 20 mA of current to ensure the small insertion loss. The vertical direction diodes (PIN Diode 2 in Figure 7) were set to OFF state with a reverse voltage of 4 V. For vertical polarization control, the PIN diode switching was then compared to the horizontal polarization control.

A gain difference of 0.7 dB was observed between the horizontal and vertical polarization states, as shown in Figure 16. The horizontal polarization had a gain of 4.2 dBi and the vertical polarization had 3.5 dBi. The half power beam width (HPBW) is approximately 30${}^{\circ }$. The cross-polarization discrimination was 17.1 dB and 16.4 dB, respectively. The gain difference is believed to be from coupling to the bias lines.

In Figure 17, the normalized peak gain at a frequency range from 2.70 GHz to 3.30 GHz is shown for a broadside pattern at 0${}^{\circ }$. The peak gain is close to the center frequency of 3 GHz at 2.995 GHz with a gain of 4.7 dB. The frequency bandwidth where gain decreases −3 dB from the peak gain is 0.36 GHz. The same plot also has the simulation results which show good agreement with the measurements around the center frequency of 3 GHz.

### 4.4. Comparison with Simulation

Then, a comparison with simulation is discussed. Initial simulation results showed a gain of 12.3 dB. However, this differs from the measured gain of 4.4 dB. A better agreement was obtained by replacing ideal conductors with copper and applying an ENIG layer as a layered impedance boundary on top of the copper traces in Ansys HFSS. The applied thicknesses were Ni 4.565 μm and Au 0.036 μm, which are based on the average measured values provided by the fabrication company. This resulted in the peak gain lowering to 6.5 dBi equaling to a difference of 2.1 dB between simulation and measurement. The reason for ENIG increasing the losses is the nickel which has a conductivity of 14.5 MS/m. For reference, the conductivity of copper is 58.0 MS/m, i.e., a four-fold difference. Even though the nickel layer is comparatively thin compared to the copper, 4.565 μm vs. 17.5 μm, the skin effect causes the current to concentrate more on the surface of the conductor, i.e., at the nickel layer. Consequently, the conduction losses increase and the radiated gain decreases. Similar observations were made in [34], where loss is reported to be 0.15 and 0.8 dB for a single antenna with a 5 μm-thick ENIG layer. For this paper, the loss is higher due to the greater number of antennas and the RF signal propagating for a relatively long distance along the feed network.

To acquire an even better agreement with measurement results, the surface roughness of the copper traces was also added. A surface roughness of 1.07 μm was applied to the layered impedance boundary which is an average value of the top and bottom side of the conductor. This approach was recommended in the documentation of HFSS as only one value for surface roughness can be added. After this addition, the peak gain became 4.5 dBi. Therefore, there is only 0.1 dB difference in the measurement results showing good agreement. The results of the comparison are shown in Figure 18. There is good agreement at the main lobe.

## 5. Discussion

### 5.1. Power Budget

To compare this work to others, a power budget is calculated for the 4×4 antenna array and the results are presented in Table 5. The ideal directivity of the aperture is calculated as $4\pi A/\lambda_{0}^{2}=15.1\;\mathrm{dBi}$ where *A* is the aperture size and $\lambda_{0}$ is the free-space wavelength. With a gain of 4.4 dBi, the aperture efficiency is 8.6%. This is the second lowest result when compared to other works of Table 6. However, it was demonstrated with simulations that the majority of the losses were from non-ideal conductors, the ENIG layer and surface roughness. Therefore, there are two simple ways to improve the gain without a redesign. First, the ENIG can be replaced with a low loss coating, e.g., organic solderability preservative (OSP) or immersion silver. This is expected to improve the gain most significantly. Second, the substrate can be replaced with a low-surface-roughness version. Substrate manufacturers commonly offer this option, as is evident from the datasheet [35]. A more laborious approach would be replacing the feed network with another design that has a shorter conduction path, e.g., a transmitarray, which has been demonstrated to reach efficiencies over 30%.

Overall, the loss sources were estimated for a horizontally polarized 0${}^{\circ }$ steering angle based on simulation results and information in a datasheet [33]. For the feed network, consisting of power dividers and phase shifter, the loss was simulated to be 0.8 dB for each design. The PIN diode has an insertion loss of around 0.4 dB at 3 GHz [33]. Without conductor losses, ENIG and surface roughness, the simulated gain of the 4×4 antenna array was 12.3 dB. Therefore, the remaining losses are estimated be from fabrication tolerance and component variations equaling 0.8 dB. The conductor and ENIG loss were simulated to be 5.8 dB in total. The remaining 2.1 dB losses were from surface roughness as demonstrated earlier.

### 5.2. Comparison with Other Works

A comparison about the related works discussed earlier in the introduction is provided in Table 6. The scan loss values in other works were picked for either E or H plane at angles 40${}^{\circ }$ or 45${}^{\circ }$ if available. Based on these values, this work is among the best performing and in general continuous control tends to perform better in this category. Interestingly, Ref. [21] has the least scan loss, even though it has discrete control. Therefore, it seems that good scan loss performance is also possible to achieve with discrete control where there is a large number of unit cells and good phase characterization. For this work, the scan loss is expected to improve if the performed curve fit has more voltage points to compare to, i.e., higher sampling. This would lead to a more accurate curve fit reducing phase error. This is especially recommended if the array size is increased further from the current 4×4 size.

As mentioned in the introduction, very few works discuss the error of the measured steering angle, as evidenced in Table 6, where only four works provide this kind of analysis. The smallest maximum error among these works is 0.8${}^{\circ }$ in [22] which is an excellent result. If the characterization in our work is improved as mentioned above, the result of this work is expected to improve from the 2${}^{\circ }$ to approximately 1${}^{\circ }$ maximum steering error. For all steering angles, this improvement would change the mean absolute error from 1.2${}^{\circ }$ to 0.8${}^{\circ }$.

The characterization method for obtaining the phase profile for beam steering is another topic that is not commonly discussed. However, this is an important aspect for obtaining a good beam steering performance, especially for designs with a continuous phase profile. In general, the characterization methods can be divided into three categories: (1) single unit cell characterization; (2) datasheet characterization; and (3) individual characterization of each unit cell. The first one has three works and the scan loss in [18,22] is higher than the works using individual characterization. Although this method is sufficient to perform beam steering, the scan loss performance would likely be improved with individual characterization or with the average model used in this work.

For papers belonging to (2), they are described as datasheet characterization as the authors only provide component values but do not describe their method. Typically, the values are similar to values provided in datasheets. However, looking at the scan loss, there is a strong variation in the results hinting that some works have a more sophisticated approach. Additionally, some of these papers only mention the varactor capacitance range which itself is not sufficient based on the experience of the authors. The reason is the lack of consideration for parasitic inductive components which can increase or decrease the total phase shift. Nevertheless, a fair comparison is not possible without more details about the methods used in these works.

Works belonging to group (3) perform the individual characterization of each unit cell. This gives consistent results but requires a lot of measurement time in an anechoic chamber. Therefore, this work proposed the phase shift characterization method, which can be performed without chamber measurements, and it captures the phase shift profile well enough to perform beam steering. However, the scan loss is expected to improve if individual characterization is performed. Additionally, the proposed method is based on calculations instead of lookup tables, meaning it can be easily readjusted or tuned. It was demonstrated in the previous section that the steering angle can be simply adjusted by increasing or decreasing the steering angle until the desired angle is achieved, demonstrating the strength of this work.

## 6. Conclusions

A re-configurable 4×4 antenna array design with electrical beam steering and polarization control was proposed. More importantly, it was demonstrated that the beam steering was adjustable. To achieve this, a characterization setup based on an average phase shift model was proposed. This was performed by measuring a partial design containing only the phase shifter and the feed network with a VNA. The benefit of this approach is that one can calculate the required phase shift with equations instead of lookup tables. This makes the beam steering adjustable as the steering is adjusted by tuning the steering angle. Additionally, good scan loss performance and accurate beam steering were obtained without individually characterizing each unit cell of the design. The proposed technique was used to steer the main beam in far-field measurements from 0${}^{\circ }$ to 45${}^{\circ }$ in 5${}^{\circ }$ steps with a maximum steering error of 2${}^{\circ }$. Additionally, the polarization control was also proven with an XPD of over 15 dB. Finally, this work also provided a loss analysis, showing that, even though the efficiency of the design was poor, it can be easily improved by choosing a different coating and substrate.

## Figures and Tables

**Figure 1 sensors-23-01324-f001:**
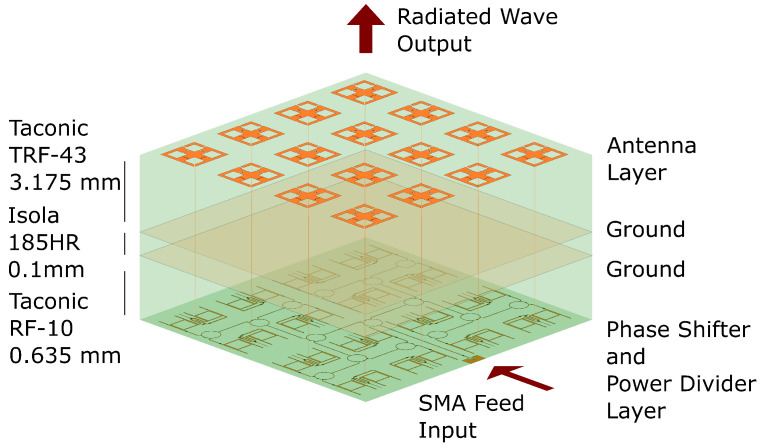
Structure of the 4×4 antenna array. The vertical dimensions are exaggerated for clarity.

**Figure 2 sensors-23-01324-f002:**
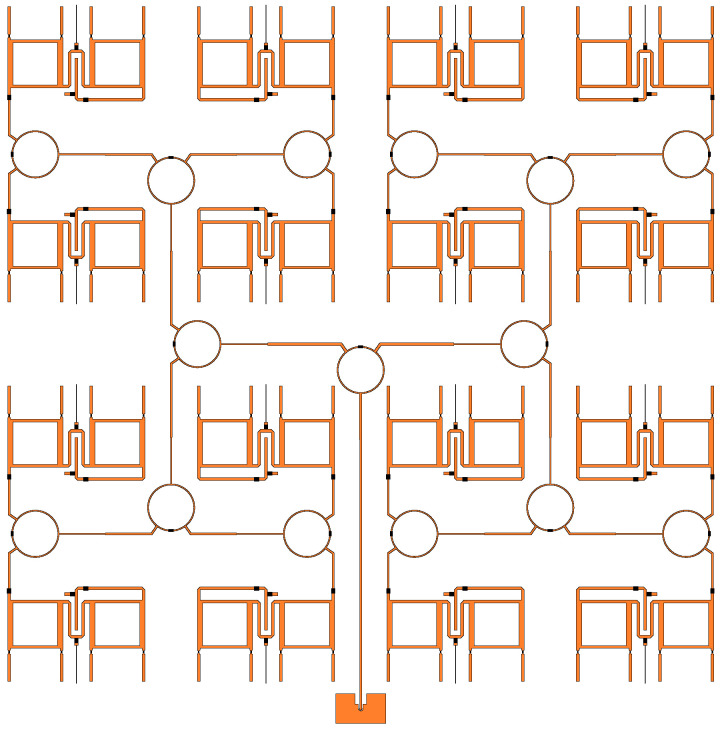
Power divider and phase shifter network.

**Figure 3 sensors-23-01324-f003:**
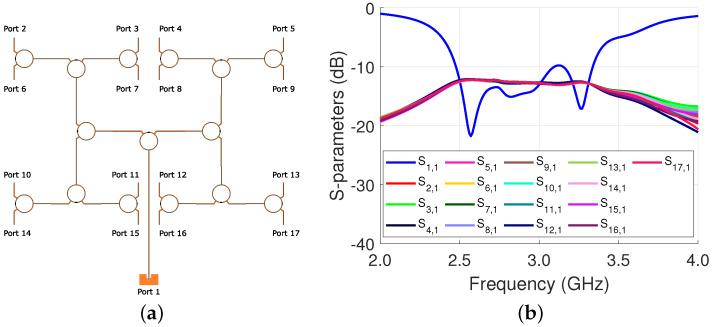
Power divider: (**a**) structure; and (**b**) simulation results.

**Figure 4 sensors-23-01324-f004:**
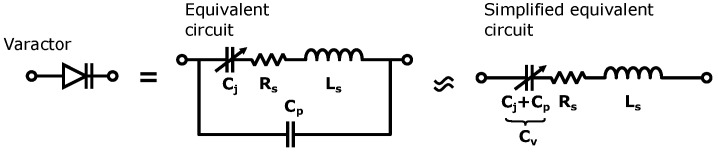
Varactor symbol and its equivalent and simplified circuit.

**Figure 5 sensors-23-01324-f005:**
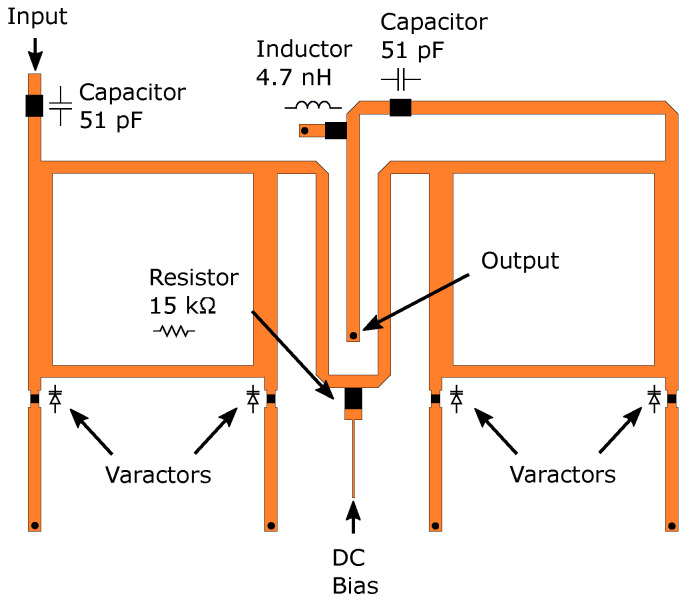
Phase shifter and its components.

**Figure 6 sensors-23-01324-f006:**
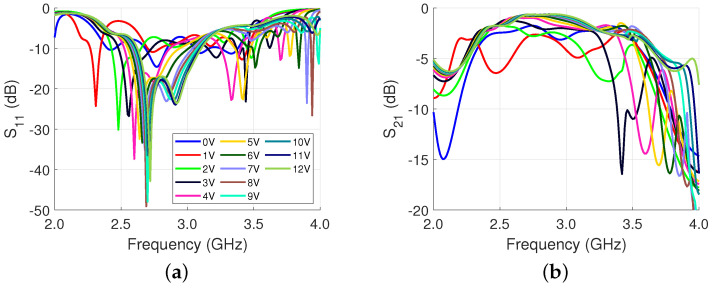
Single phase shifter simulation results when the voltage is swept from 0 V to 12 V: (**a**) $S_{11}$; and (**b**) $S_{21}$.

**Figure 7 sensors-23-01324-f007:**
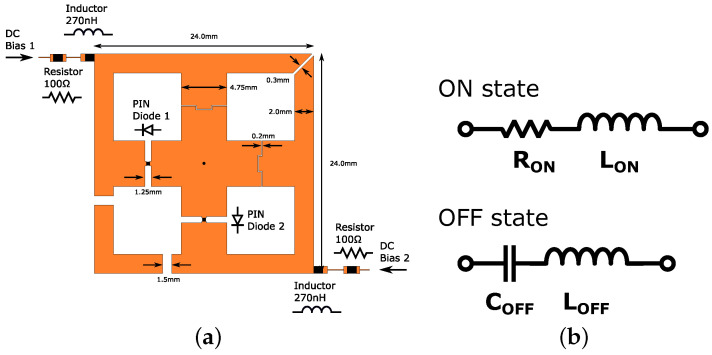
Antenna with dual polarization control through two individually controlled PIN diodes: (**a**) antenna; and (**b**) equivalent circuit of PIN diode in ON and OFF state.

**Figure 8 sensors-23-01324-f008:**
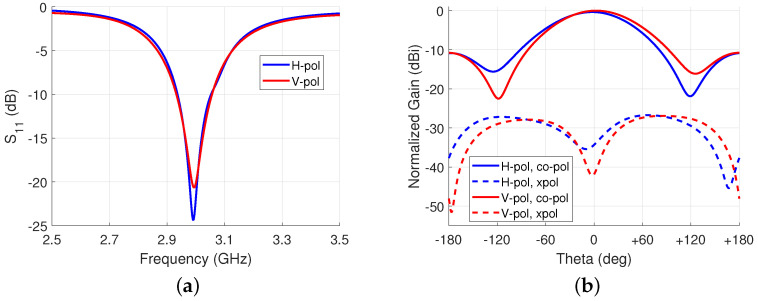
Simulation results of a single antenna at broadside: (**a**) $S_{11}$ for both polarizations; and (**b**) normalized gain at 3 GHz for both polarizations.

**Figure 9 sensors-23-01324-f009:**
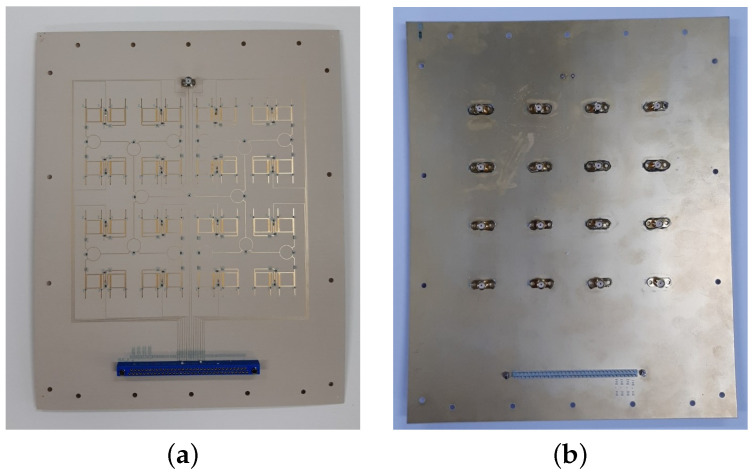
Fabricated a 4×4 structure without an antenna for phase shift characterization: (**a**) front side; and (**b**) back side.

**Figure 10 sensors-23-01324-f010:**
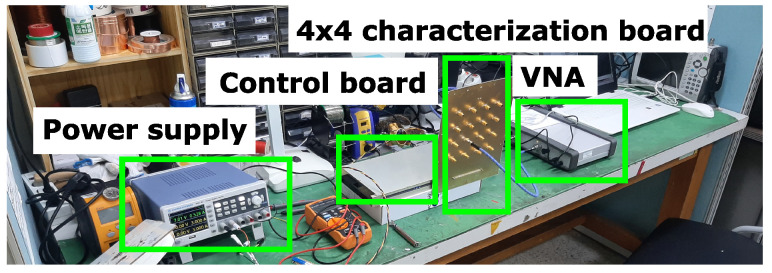
Measurement setup for determining the phase shift profile.

**Figure 11 sensors-23-01324-f011:**
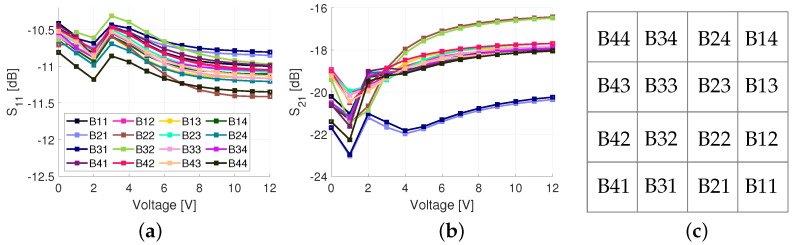
Measurement results of the 4×4 structure without an antenna at 3 GHz when varactors are biased from 0 to 12 V: (**a**) $S_{11}$ measurement; (**b**) $S_{21}$ measurement; and (**c**) unit cell mapping of the 4×4 antenna array.

**Figure 12 sensors-23-01324-f012:**
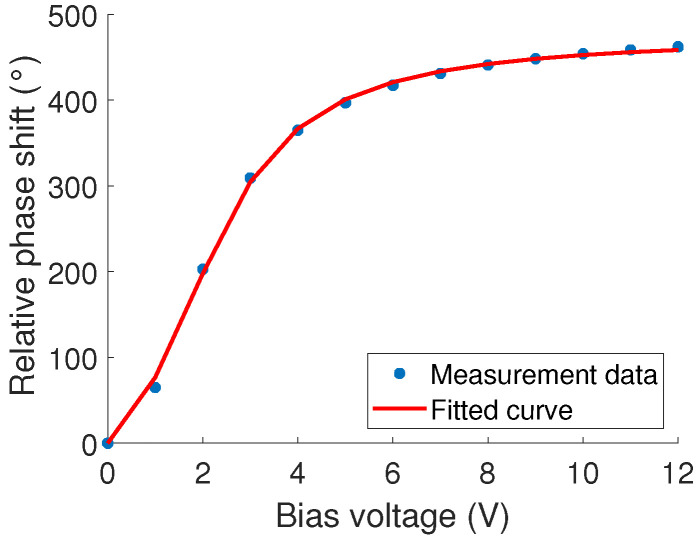
Curve fit result based on the measured average phase shift.

**Figure 13 sensors-23-01324-f013:**
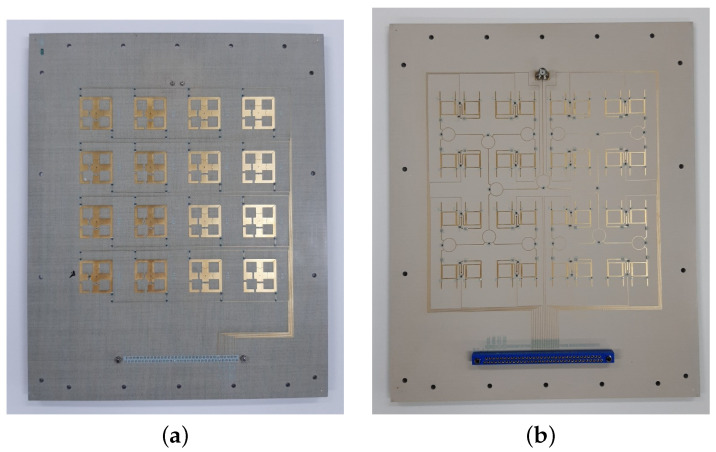
Fabricated 4×4 structure: (**a**) antenna side; (**b**) phase shifter and power divider side.

**Figure 14 sensors-23-01324-f014:**
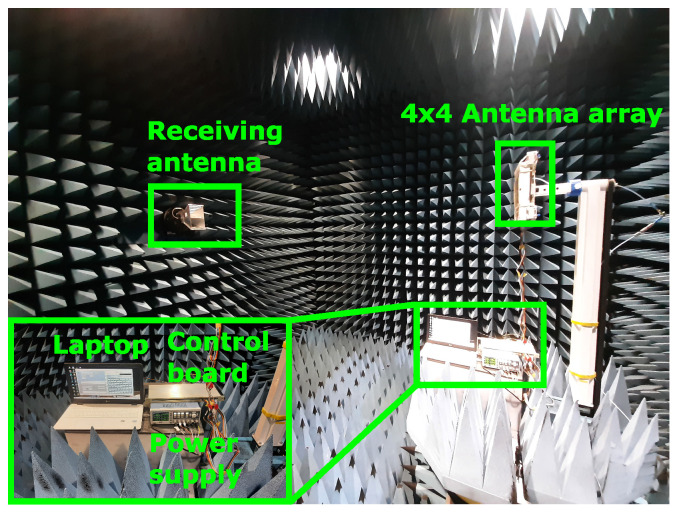
Measurement setup for far-field measurements in the anechoic chamber.

**Figure 15 sensors-23-01324-f015:**
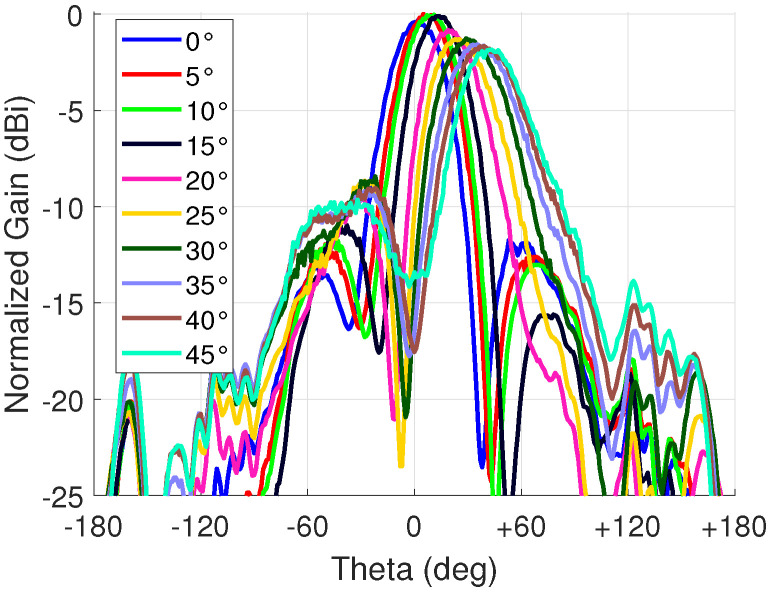
Beam steering far-field radiation patterns at 3 GHz.

**Figure 16 sensors-23-01324-f016:**
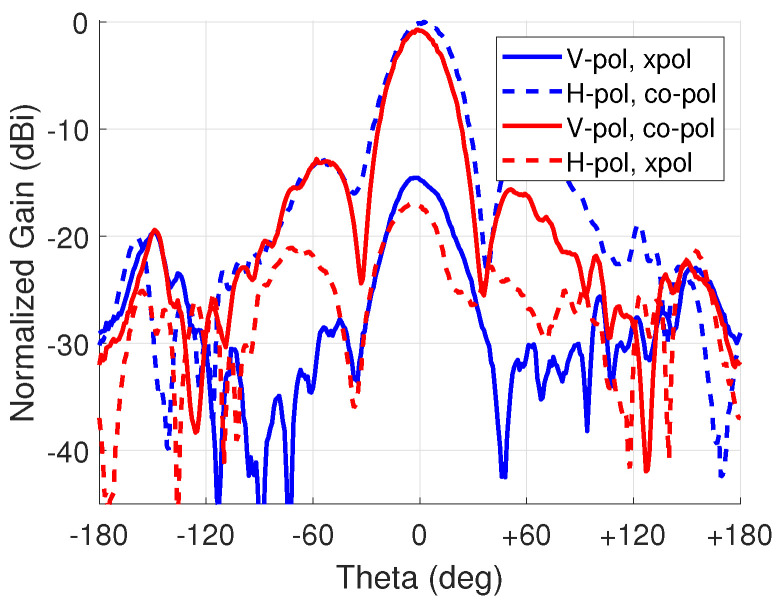
Polarization control results for vertical and horizontal polarization, respectively.

**Figure 17 sensors-23-01324-f017:**
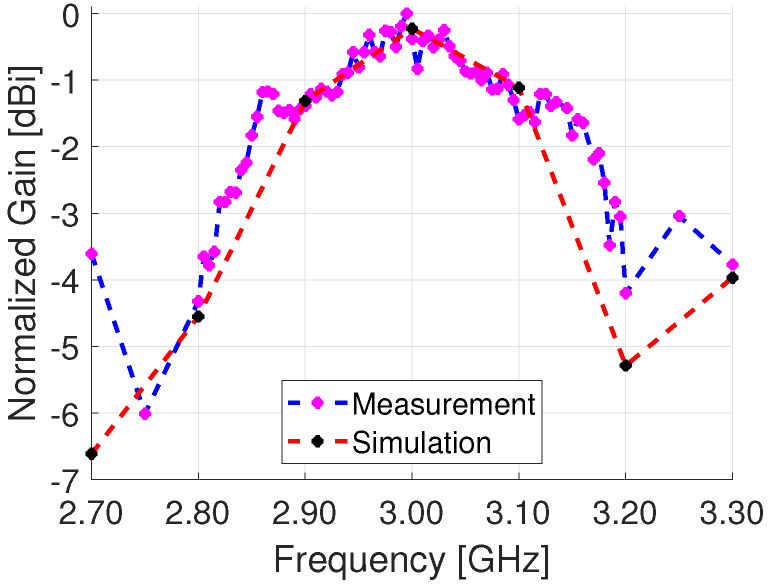
Radiated peak gain from 2.7 GHz to 3.3 GHz.

**Figure 18 sensors-23-01324-f018:**
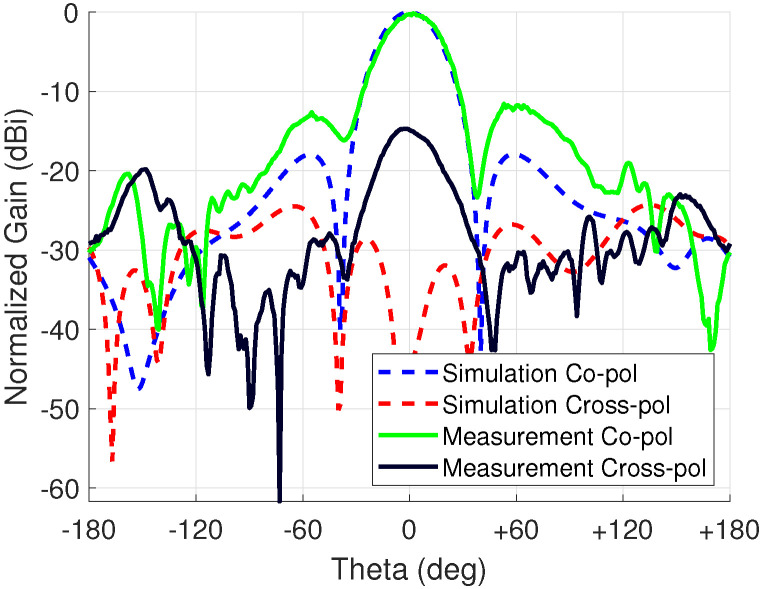
Measurement comparison with simulation at 3 GHz.

**Table 1 sensors-23-01324-t001:** Varactor parameters obtained from curve fit.

Parameter	Value
$L_{S}$	1.23 nH
$C_{\mathrm{jo}}$	1.23 pF
$V_{j}$	1.92 V
*m*	2.06
$C_{P}$	0.25 pF

**Table 2 sensors-23-01324-t002:** Phase measurements to determine the amount of phase error in different unit cells for 10${}^{\circ }$ beam steering case.

Row 3					Row 4				
**Unit Cell**	**Calculated**	**Measured**	**Voltage**	**Error**	**Unit Cell**	**Calculated**	**Measured**	**Voltage**	**Error**
B12	0.0${}^{\circ }$	0.0${}^{\circ }$	4.4 V	0.0${}^{\circ }$	B11	0.0${}^{\circ }$	0.0${}^{\circ }$	4.4 V	0.0${}^{\circ }$
B22	25.0${}^{\circ }$	23.3${}^{\circ }$	5.3 V	1.7${}^{\circ }$	B21	25.0${}^{\circ }$	25.3${}^{\circ }$	5.3 V	−0.3${}^{\circ }$
B32	50.0${}^{\circ }$	49.2${}^{\circ }$	7.0 V	0.8${}^{\circ }$	B31	50.0${}^{\circ }$	50.3${}^{\circ }$	7.0 V	−0.3${}^{\circ }$
B42	75.0${}^{\circ }$	75.0${}^{\circ }$	12.0 V	0.0${}^{\circ }$	B42	75.0${}^{\circ }$	79.7${}^{\circ }$	12.0 V	−4.6${}^{\circ }$

**Table 3 sensors-23-01324-t003:** Phase measurements to determine the amount of phase error in different unit cells for 40${}^{\circ }$ beam steering case.

Row 3					Row 4				
**Unit Cell**	**Calculated**	**Measured**	**Voltage**	**Error**	**Unit Cell**	**Calculated**	**Measured**	**Voltage**	**Error**
B12	0.0${}^{\circ }$	0.0${}^{\circ }$	1.9 V	0.0${}^{\circ }$	B11	0.0${}^{\circ }$	0.0${}^{\circ }$	1.9 V	0.0${}^{\circ }$
B22	92.6${}^{\circ }$	97.7${}^{\circ }$	2.7 V	−5.2${}^{\circ }$	B21	92.6${}^{\circ }$	94.5${}^{\circ }$	2.7 V	−1.9${}^{\circ }$
B32	185.1${}^{\circ }$	174.9${}^{\circ }$	4.0 V	10.2${}^{\circ }$	B31	185.1${}^{\circ }$	168.8${}^{\circ }$	4.0 V	16.3${}^{\circ }$
B42	277.7${}^{\circ }$	270.5${}^{\circ }$	12.0 V	7.2${}^{\circ }$	B42	277.7${}^{\circ }$	269.0${}^{\circ }$	12.0 V	8.7${}^{\circ }$

**Table 4 sensors-23-01324-t004:** Beam steering results at 3 GHz for the horizontally polarized wave.

Target Angle	Measured Angle	Angle Error	Control Angle	Peak Gain	SLL
$0^{\circ }$	$2^{\circ }$	$+2^{\circ }$	$0^{\circ }$	4.4 dBi	−11.4 dB
$5^{\circ }$	$5^{\circ }$	$0^{\circ }$	$3^{\circ }$	4.7 dBi	−12.0 dB
$10^{\circ }$	$11^{\circ }$	$+1^{\circ }$	$5^{\circ }$	4.6 dBi	−11.4 dB
$15^{\circ }$	$13^{\circ }$	$-2^{\circ }$	$10^{\circ }$	4.6 dBi	−10.6 dB
$20^{\circ }$	$19^{\circ }$	$-1^{\circ }$	$20^{\circ }$	3.8 dBi	−8.5 dB
$25^{\circ }$	$23^{\circ }$	$-2^{\circ }$	$30^{\circ }$	3.4 dBi	−7.4 dB
$30^{\circ }$	$29^{\circ }$	$-1^{\circ }$	$35^{\circ }$	3.4 dBi	−7.2 dB
$35^{\circ }$	$35^{\circ }$	$0^{\circ }$	$40^{\circ }$	3.1 dBi	−7.6 dB
$40^{\circ }$	$39^{\circ }$	$-1^{\circ }$	$44^{\circ }$	3.0 dBi	−7.4 dB
$45^{\circ }$	$43^{\circ }$	$-2^{\circ }$	$45^{\circ }$	2.8 dBi	−7.8 dB

**Table 5 sensors-23-01324-t005:** Power budget at 3 GHz for a broadside horizontally polarized beam.

Parameter	Value
Aperture size	160 mm × 160 mm
Ideal Directivity	15.1 dBi
Measured gain	4.4 dBi
Feed network loss	0.8 dB
Phase shifter loss	0.8 dB
PIN diode loss	0.4 dB
Fabrication tolerance losses	0.8 dB
ENIG and conductor loss	5.8 dB
Surface roughness	2.1 dB

**Table 6 sensors-23-01324-t006:** Comparison table of this work to other works.

Reference	Peak Gain	Steered Gain (Angle)	Scan Loss	Max Steering Error	Phase Profile	Characterization	Frequency	Array Size	Aperture Efficiency
[18]	20.8 dBi	18.3 dBi (-40${}^{\circ }$)	2.5 dB	N/A	Discrete	Single, waveguide	29 GHz	20 × 20	9.5%
[14]	17.0 dBi	14.9 dBi (40${}^{\circ }$)	2.1 dB	N/A	Continuous	Datasheet ^1^	5.4 GHz	8 × 8	28.5%
[23]	14.6 dBi	12.9 dBi (45${}^{\circ }$)	1.9 dB	N/A	Continuous & Discrete	Datasheet ^1^	4.8 GHz	6 × 6	27.6%
[15]	15.0 dBi	13.4 dBi (40${}^{\circ }$)	1.6 dB	N/A	Continuous	Individual, method unclear	5.0 GHz	6 × 6	28.2%
[24]	13.8 dBi	N/A (60${}^{\circ }$)	2.8 dB	0.9${}^{\circ }$	Discrete	Method unclear	14.8 GHz	24 × 2	6.3%
[19]	16.8 dBi	14.5 dB (40${}^{\circ }$)	2.3 dB	N/A	Discrete	Datasheet ^1^	5.0 GHz	16 × 16	18.4%
[16]	15.6 dBi	13.4 dB (45${}^{\circ }$)	2.2 dB	1.9${}^{\circ }$	Continuous	Individual, near field probe	4.8 GHz	6 × 6	34.0%
[17]	19.9 dBi	N/A	N/A	16${}^{\circ }$	Continuous	Single unit cell	5.2 GHz	5 × 5	N/A
[25]	23.7 dBi	20.0 dBi (−45${}^{\circ }$)	3.7 dB	N/A	Continuous and discrete	Datasheet ^1^	5.6 GHz	16 × 16	33.5%
[20]	21.3 dBi	15.4 dBi (60${}^{\circ }$)	5.9 dB	N/A	Discrete	Datasheet ^1^	28 GHz	20 × 20	12.5%
[21]	21.4 dBi	19.9 dBi (40${}^{\circ }$)	1.5 dB	N/A	Discrete	Datasheet ^1^	13.5 GHz	16 × 16	14.8%
[22]	13.4 dBi	10.7 dBi (45${}^{\circ }$)	2.7 dB	0.8${}^{\circ }$	Discrete	Single, PIN & phase shifter measurement	12 GHz	16 × 2	24.5%
This work	4.7 dBi	3.0 dBi (40${}^{\circ }$)	1.7 dB	2${}^{\circ }$	Continuous	Measurement, average phase shift	3 GHz	4×4	8.6%
		2.8 dBi (45${}^{\circ }$)	1.9 dB						

^1^ Works that only present component values for PIN diodes and/or varactors are marked as datasheet.

## Data Availability

Not applicable.

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
