# Peer review of "A 4 × 4 Active Antenna Array with Adjustable Beam Steering"

_sensors, 2023, doi:10.3390/s23031324_

Round 1

Reviewer 2 Report

In this study, the electronic beam steering of the antenna was carried out . The expected and actual angles of beam were presented and compared. The error was revealed. The working frequency of system was presented as 3GHz. The reviews are presented below.

1. What is used as control board? What is the circuit diagram of your control structure?

2. Which protocol is employed for communication between pc and control board.

3. Line 432: The results are reported as "from the 2â—¦ to around 1â—¦ steering error". Instead of that, an error metric (like MAE or RMSE or etc.) should be employed.

Round 2

Reviewer 2 Report

In line with the criticisms, changes were made in the study and corrections were made. I think that this study is more interesting, understandable and comparable. The study is eligible for publication as it is.